# Characteristics of Subtype and Molecular Transmission Networks among Newly Diagnosed HIV-1 Infections in Patients Residing in Taiyuan City, Shanxi Province, China, from 2021 to 2023

**DOI:** 10.3390/v16071174

**Published:** 2024-07-22

**Authors:** Ruihong Gao, Wentong Li, Jihong Xu, Jiane Guo, Rui Wang, Shuting Zhang, Xiaonan Zheng, Jitao Wang

**Affiliations:** 1Academy of Medical Sciences, Shanxi Medical University, 56 Xinjian South Road, Taiyuan 030001, Shanxi, China; liwt502@163.com; 2School of Public Health, Shanxi Medical University, 56 Xinjian South Road, Taiyuan 030001, Shanxi, China; 3Taiyuan Center for Disease Control and Prevention, No. 22, Huazhang West Street, Xiaodian District, Taiyuan 030012, Shanxi, China; zyzd919@163.com (J.X.); jianeg1@163.com (J.G.); wangrui871020@163.com (R.W.); 18834185181@163.com (S.Z.); 18018952836@163.com (X.Z.)

**Keywords:** HIV-1, subtype, cluster, transmission network, molecular epidemiology

## Abstract

The HIV-1 pandemic, spanning four decades, presents a significant challenge to global public health. This study aimed to understand the molecular transmission characteristics of newly reported HIV infections in Taiyuan, Shanxi Province, China, to analyze the characteristics of subtypes and the risk factors of the transmission network, providing a scientific basis for precise prevention and intervention measures. A total of 720 samples were collected from newly diagnosed HIV-1 patients residing in Taiyuan between 2021 and 2023. Sequencing of partial genes of the HIV-1 *pol* gene resulted in multiple sequence acquisitions and was conducted to analyze their subtypes and molecular transmission networks. Out of the samples, 584 *pol* sequences were obtained, revealing 17 HIV-1 subtypes, with CRF07_BC (48.29%), CRF01_AE (31.34%), and CRF79_0107 (7.19%) being the dominant subtypes. Using a genetic distance threshold of 1.5%, 49 molecular transmission clusters were generated from the 313 *pol* gene sequences. Univariate analysis showed significant differences in the HIV transmission molecular network in terms of HIV subtype and household registration (*p* < 0.05). Multivariate logistic regression analysis showed that CRF79_0107 subtype and its migrants were associated with higher proportions of sequences in the HIV transmission network. These findings provide a scientific foundation for the development of localized HIV-specific intervention strategies.

## 1. Introduction

Human immunodeficiency virus (HIV) has been a serious threat to global public health and disease surveillance systems for the past 40 years [1,2,3,4]. HIV can be categorized into two primary types: HIV-1, which is the primary causative agent for the dissemination of acquired immunodeficiency syndrome (AIDS), and HIV-2 [5]. Furthermore, HIV-1 is widespread globally and is the primary type of infection in China. In contrast, HIV-2 is primarily prevalent in countries in West Asia [6]. Among the four phylogeny groups that constitute HIV-1 (M, N, O, P), the M group is considered the primary virulence factor in the AIDS pandemic [7]. The M group includes nine major subtypes (i.e., A–D, F–H, J, and K), over 110 circulating recombinant forms (CRFs), and numerous unique recombinant forms (URFs) [8]. Due to genetic variation and cross-recombination of genes between different genotypes, numerous new CRFs and unique recombinant forms (URFs) are continually emerging, significantly altering the global epidemic landscape of HIV. According to the World Health Organization (https://www.who.int/ (accessed on 6 April 2024)), the estimated number of persons living with HIV/AIDS (PLWH) globally by the end of 2022 was 39 million, with 1.3 million new infections and approximately 630,000 people dying annually from AIDS-related illnesses. Despite the implementation of the 9‘0-90-90’ prevention strategy proposed by the Joint United Nations Program on HIV/AIDS (UNAIDS) [9,10], China continues to face a high prevalence of HIV-1 subtypes, presenting a complex and diverse challenge with significant obstacles and pressures [11].

Since the discovery of the first HIV-1 infection in China in 1985 [12], the number of PLWH has been increasing year after year. By the end of 2023, the number of PLWH in China was 1,289,700, with 110,491 newly diagnosed in 2023 [13]. The distributional characteristics and epidemiological trends of HIV-1 subtypes are experiencing significant changes concurrent with the rapid rise in the number of PLWH [11]. Four national HIV molecular epidemiological surveys have been conducted; the latest in 2016, showed a rapid increase in HIV-1 subtypes [14]. More than 20 HIV-1 subtypes have been identified, with CRF07_BC, CRF01_AE, CRF08_BC, and B being the most prevalent epidemic subtypes [15]. Since the 1990s, CRF07_BC and CRF01_AE have continuously evolved in China, leading to a widespread epidemic among various risk groups, such as men who have sex with men (MSM), commercial heterosexual individuals, and people who inject drugs (PWID) [16,17,18,19,20,21]. Newly reported circulating recombinant form strains (CRFs) and unique recombinant form strains (URFs) are frequently reported within specific groups [22,23,24,25].

The molecular networks utilize the genetic similarities between viral sequences to achieve the spread of the virus. Previously, researchers relied primarily on questionnaires and partner tracing to construct social or sexual transmission networks. Subsequently, they utilized genetic information from HIV sequences to develop molecular transmission networks [26,27,28]. Zhang et al. employed molecular networks to quantify HIV transmission patterns in Hangzhou, thereby offering local evidence for the development of precise HIV prevention strategies [29]. He et al. used molecular networks to analyze HIV-1 transmission clusters in Guangxi, identifying distinct transmission networks and clusters [30]. The utilization of molecular networks for constructing a macro-propagation network for PLWH is more precise and has been extensively used to validate the findings and conclusions of epidemiological field investigations. This approach can effectively mitigate information disparities and enhance the credibility of conclusions [31,32,33].

Taiyuan City, comprising 10 counties (districts/cities), is the capital of Shanxi Province, China. Positioned strategically in the heart of Shanxi Province, Taiyuan City serves as a significant transportation hub in northern China, attracting a large population flow. To gain a comprehensive understanding of the distribution and characteristics of HIV-1 subtypes in Taiyuan, we utilized phylogenetic inference, transmission network analysis, and molecular epidemiological data to investigate newly diagnosed HIV-1 infections between 2021 and 2023.

## 2. Materials and Methods

### 2.1. Population Study and Sample Collection

Patients newly diagnosed with HIV-1 infection, residing in Taiyuan between 2021 and 2023, were included in the study. The samples were selected based on the following criteria: (1) serum or plasma samples collected by the Taiyuan Center for Disease Control and Prevention (Taiyuan CDC) testing positive for HIV-1 using Western blot testing; (2) participants must reside, work, or be employed in Taiyuan, regardless of their household registration address; and (3) the household registration address is determined based on the information from the ID card. The epidemiological data (including, gender, ethnicity, age, marital status, education background, occupation, transmission route, etc.) were acquired from the China Information System for Disease Control and Prevention.

### 2.2. HIV-1 RNA Extraction, Amplification, and Sequencing

Viral RNA was extracted from specimens using a MagMAX™-96 Viral RNA Isolation Kit (Thermo Fisher Scientific, Foster City, CA, USA), according to the manufacturer’s instructions. RNA was eluted with 50 µL of elution buffer and used immediately. Reverse transcriptase–polymerase chain reaction (RT-PCR) and nested PCR were used to amplify the HIV-1 partial polymerase (*pol*) gene (1.3 kp, HXB2:2253–3553). PrimeScript^TM^ One Step Reverse Transcriptase–Polymerase Chain Reaction Kit Version 2 (Takara, Dalian, China) and the Ex Taq Kit (Takara, Dalian, China) were employed in the amplification. The primer sequences and thermal cycling conditions were described previously [34]. The PCR products were analyzed using 1% agarose gel electrophoresis. The target PCR products were sent to Tsingke Biotechnology Co, Ltd. for purification and sequencing using an ABI 3730XL DNA sequencer (Applied Biosystems, Carlsbad, CA, USA), with five overlapping primers (https://www.chinacdc.cn/ (accessed on 15 June 2023)).

### 2.3. HIV-1 Genotyping, Phylogenetic Analysis, and Amino Acid Difference Analysis

Sequences were assembled using Sequencher 5.4.6 (Gene Codes, AnnArbor, MI). BioEdit (version 7.0.9) software (http://www.mbio.ncsu.edu/BioEdit/ (accessed on 15 February 2024)) was used for sequence alignment. HIV-1 subtypes were determined using *pol* sequences by the Los Alamos National Laboratory HIV Database (http://www.hiv.lanl.gov (accessed on 25 February 2024)) and subsequently clarified by phylogenetic analysis. Neighbor-joining phylogenetic trees were constructed using the Kimura two-parameter model with 1000 bootstrap replicates in MEGA software (Version 11.0.11); the check value was 70% to identify the subtype and to control for potential laboratory and sample contamination [35]. We used the function of computing pairwise distance (P-distance) and determined average within-group and between-group distance in MEGA 11.0.11 in order to distinguish differences among all subtypes. Subtype reference alignments were downloaded from HIV databases (www.hiv.lanl.gov (accessed on 25 February 2024)), including 41 HIV-1 group M subtypes and 45 common CRFs, at home and abroad.

### 2.4. Molecular Transmission Network Analysis

The pairwise genetic distances were estimated using the Tamura-Nei 93 (TN93) fast distance calculator, and the HIV molecular network was created using the HIV-TRACE tool, with a threshold genetic distance of 0.015 among HIV-1 subtypes [36]. All nodes in the HIV molecular network were assigned with epidemiological data, and molecular network maps were generated. In this study, clusters with five or more nodes were defined as larger clusters. Utilizing the connections among various risk behavior groups within the network, we constructed Sankey diagrams to depict the links of different HIV-1 subtypes among the various risk groups in the network.

### 2.5. Statistical Analysis

All data were entered into Microsoft Excel 2010 (Microsoft Corporation; Redmond, WA, USA). Statistical analysis was conducted using SPSS 26.0 (SPSS, Inc; Chicago, IL, USA). Statistical comparisons were performed using Fisher’s exact test, Chi-square testing, and multivariate logistic regression, which are used to identify the influencing factors that are associated with inclusion in the clusters in molecular transmission networks among the participants. A *p*-value less than 0.05 (typically ≤0.05) is statistically significant.

### 2.6. Ethical Statement

This study was approved by the Research Ethics Review Committee of the Taiyuan CDC (Approval ID: 2023020). All procedures were performed following the guidelines of the Declaration of Helsinki, as well as international and national laws, regulations, and guidelines for human studies.

## 3. Results

### 3.1. Epidemiological Information of the Study Subjects

We enrolled 720 individuals newly diagnosed with HIV-1 from 2021 to 2023 in Taiyuan in the study, and we successfully obtained and analyzed *pol* sequences from 584 (81.1%, 584/720) samples. The epidemiological information is summarized in Table 1. The mean age was 37 years, ranging from 10 to 85 years. Most of the participants were male (90.7%, 653/720), single (49.2%, 354/720), of Han ethnicity (98.8%, 711/720), had achieved education beyond the compulsory level (63.7%, 459/720), and belonged to registered households in Taiyuan (63.8%,459/720). The major route of infection was heterosexual transmission (62.1%, 447/720), followed by homosexual transmission (36.3%, 261/720), and other routes of transmission accounted for 1.7% (12/720), which includes mother-to-child transmission (MTCT) and injection drug use (IDU). The top three districts identified in the study were Xiaodian, Wanbailin, and Xinghualing, representing 27.1% (195/720), 22.5% (162/720), and 15.1% (109/720) of cases, respectively. Moreover, the largest number of samples was collected in 2022 (282 samples), followed by the 2023 (270 samples) and the 2021 periods (168 samples).

### 3.2. HIV-1 Subtype Distribution and Phylogenetic Analysis and Amino Acid Difference Analysis

A total of 17 HIV-1 subtypes and a proportion of URFs were identified among the 584 successfully obtained *pol* sequences. The distribution of these subtypes is depicted in Figure 1. CRF07_BC (282, 48.29%), CRF01_AE (183, 31.34%), and CRF79_0107 (42, 7.19%) emerged as the major subtypes. Subsequently, B (22, 3.77%), CRF55_01B (10, 1.71%), URF (10, 1.71%), CRF08_BC (7, 1.20%), CRF107_01B (5, 0.86%), CRF80_0107 (5, 0.86%), CRF103_01B (4, 0.68%), CRF114_0155 (4, 0.68%), CRF113_0107 (3, 0.51%), CRF115_01C (2, 0.34%), CRF67_01B (2, 0.34%), C (1, 0.17%), CRF104_0107 (1, 0.17%), and CRF76_01B (1, 0.17%) were also identified.

We further compared the sequences of all subtypes with reference sequences from the whole country to clarify the phylogenetic relationship between HIV-1 subtypes in Taiyuan and the other regions. As shown in Figure 2, the strains from Taiyuan were distributed in multiple clades in the neighbor-joining trees, with some forming a local cluster within the Taiyuan area, and immigrants and locals are not segregated into distinct clusters, but are intertwined.

Since that CRF07_BC and CRF01_AE are the top two dominant strains in Taiyuan, we further aligned our sequences of CRF07_BC and CRF01_AE subtypes with the reference sequences. CRF07_BC displays two distinct evolutionary branches, as depicted in Appendix A. Among these, the CRF07_BC_N population accounted for a substantial 69.15% (195/282); meanwhile, the CRF07_BC_O population accounted for 30.85% (87/282). As shown in Appendix A, theCRF01_AE-infected individuals exhibited relatively high proportions of C4 and C5 subclusters, comprising 63.4% (116/183) and 21.9% (40/183), respectively. Furthermore, within the CRF01_AE subtype, we identified additional subclusters, specifically C1, C2, and C3, as well as a distinct O subcluster.

The overall distribution characteristics of epidemic subtypes in Taiyuan are shown in Table 2. Gender and education differences among subtypes were the statistically significant factors (*p* < 0.05). The infection primarily affected males, accounting for 90.6% (529/584). CRF07_BC primarily infected individuals aged 50 and above, accounting for 57.1% (97/170) of the age groups. Similarly, those with a college education or higher also exhibit a high prevalence of the CRF07_BC subtype, accounting for 46.5% (114/245). Compared to local individuals, migrants exhibited a higher proportion of other subtypes. Heterosexual transmission remained the primary transmission route, accounting for 49.0% (179/365) of CRF07_BC-infected individuals and 30.7% (112/365) of CRF01_AE-infected individuals.

In addition, Figure 3 presents the trend analysis of each subtype from 2021 to 2023, along with the research findings regarding the trend of subtype changes among migrants and locals during the same period. Interesting trends emerge from a closer inspection of the data. In 2021, the prevalence of CRF07_BC increased among the study participants and migrants, yet local CRF07_BC and CRF01_AE proportions were stable. Subsequently, from 2022 to 2023, there was a rise in CRF07_BC observed in both the local and overall data. The CRF07_BC trend among immigrants consistently reflected this overall trend.

We calculated the average differences in amino acids and nucleotides among various subtypes, as well as within them (Appendix A). The results indicate that the average nucleotide differences among HIV-1 subtypes in Taiyuan from 2021 to 2023 ranged from 0.4% to 7.4%, with the URF subtype showing even larger differences, surpassing the average significantly. Likewise, the average difference in amino acids among the URF subtype also greatly exceeds the norm.

### 3.3. HIV-1 Molecular Transmission Network

At a genetic distance of a 1.5% gene distance threshold, the transmission cluster analysis identified 313 individuals (43.5%, 313/720) within the transmission network, revealing 50 HIV transmission clusters with sizes ranging from 2 to 133 nodes. A logistic regression model was used to analyze the influencing factor of a node degree ≥ 2 on the network. We found that compared to locals, individuals who were migrants had significantly higher odds of clustering (adjusted odds ratio [AOR] 2.124; 95% confidence interval: 1.457–3.095, *p* < 0.001). Individuals within clusters had a higher likelihood of belonging to the subtype CRF79_0107 (AOR: 4.270 [2.487–7.334], *p* < 0.001). See Table 3 for details.

Among these clusters, the CRF07_BC network exhibited a higher number of newly diagnosed HIV-1 infections compared to the CRF01_AE network, with the latter forming multiple smaller clusters (Figure 4). Among the nine clusters with five or more nodes detailed in Table 4, four clusters were associated with CRF07_BC, two with CRF01_AE, and one cluster each with CRF79_0107, CRF107_01B, and URF. Of the 219 individuals within the nine larger clusters, the majority (64.4%, 141/219) were infected through heterosexual transmission. The sequences of CRF07_BC formed the largest HIV-1 transmission cluster, CRF07_BC_Cluster1, comprising 58 MSM, 73 heterosexual individuals, and 2 IDUs. The correlation among MSM within this cluster was the highest. CRF07_BC_Cluster3 and CRF07_BC_Cluster8 exhibited a preponderance of elderly individuals, with heterosexual transmission being the primary route. Furthermore, for CRF79_0107_Cluster2, the molecular network analysis revealed a significant cluster comprising 27 individuals. Among these individuals, 19 were locals, while 8 were migrants. Moreover, the two CRF01_AE clusters, Cluster5 and Cluster6, each comprised eight individuals, all of whom were heterosexual males. In addition to these, we also identified CRF107_01B, mostly comprised of local males, with 40% (2/5) and 60% (3/5) heterosexual and homosexual transmission, respectively.

From the data in Figure 5, it is apparent that in the CRF07_BC network (which included 965 links), the correlation between heterosexual transmission within the same risk group was the highest, accounting for 44.5% (429/965); among all cross-risk groups, MSM had the highest correlation with the heterosexual group, accounting for 18.92% (200/965). We delved into the characteristics of the HIV-1 subtypes among these different risk groups within the network. The CRF01_AE network and the other subtypes network show the same relationship between the various risk groups in the network.

## 4. Discussion

In this study, we collected samples of newly diagnosed HIV-1 infections from patients residing in Taiyuan from 2021 to 2023. Subsequently, we carried out a detailed molecular epidemiological study, which included analyses of phylogenetics, transmission characteristics, and risk factors. The objective of this study was to investigate newly diagnosed HIV-1 infections and conduct a comprehensive analysis of the local transmission characteristics and epidemic patterns of HIV-1 subtypes across various risk groups.

Most HIV-1 infections in Taiyuan are diagnosed in male patients, which is consistent with the results of studies conducted in other provinces of China [37,38]. Over the years, there has been a gradual increase in female infections, with heterosexual sex being the main transmission route. Epidemiological data indicate that sexual transmission has emerged as the primary route of HIV-1 transmission, bringing the disease to the general population and presenting a significant challenge to HIV/AIDS prevention and control in China [21,39,40,41]. Our study finds that the majority of new HIV-1 infections occur in individuals aged 50 years or above, a proportion that appears to be higher than that noted in previous studies [42]. Newly infected individuals aged 50 and above account for 27.8% (200/720), ranking first among the age groups; this result is likely attributed to the lack of self-protection awareness among the elderly, as there is a low rate of condom use and a limited level of knowledge about AIDS among this age group. This situation can potentially lead to high-risk sexual behavior, thereby further increasing the risk of infection and transmission within the elderly population [43,44]. Targeted health education and prevention interventions tailored to the behavioral and psychological characteristics of the elderly need to be developed. On the other hand, individuals under 30 years old account for a significant proportion of infections (24.7%, 178/720), with a majority of infections occurring through homosexual transmission; this age group is sexually active and also represents the key population to target for preventing HIV spread.

A total of 17 HIV-1 subtypes were identified. Among the detected subtypes, CRF07_BC represented the predominant subtype. This finding significantly differed from those of the 2016–2017 survey and those from other regions, where CRF01_AE (i.e., Anhui, Liaoning and Guangxi), B (i.e., Henan), and CRF08_BC (i.e., Yunnan) were the dominant subtypes [40,42,45,46,47]. CRF07_BC_N is primarily transmitted through heterosexual behavior, whereas CRF07_BC_N, previously prevalent among MSM, has now shifted its main route of transmission [48]. Additionally, CRF07_BC_O was previously primarily transmitted among PWID and heterosexual individuals. In this study, it continued to be primarily transmitted among heterosexuals, aligning with the finding from previous research [48]. These findings indicate that this subtype has broadened its transmission to include individuals at risk of sexually transmitted infections. The second subtype, CRF01_AE, has experienced a decrease in proportion compared to the rates noted in previous studies, with different subtypes now emerging. CRF01_AE remains the predominant subtype of HIV-1 in Taiyuan during 2016–2017, and is particularly prevalent among MSM, specifically the C4 and C5 subtypes [42]. Recent research indicates that the CRF01_AE subtype continues to be predominantly composed of the C4 and C5 subclusters, followed by the C1, C2, C3, and O subclusters, which are more commonly transmitted by MSM. One unanticipated result of this study is the non-detection of the subtype CRF65_cpx, which was previously found in the MSM population in Taiyuan [49]; this absence suggests that this particular subtype may be a susceptible strain in Taiyuan and could be eradicated through ongoing evolution. Another possibility is that samples of this subtype are not successfully amplified.

However, the phylogenetic tree did not effectively illustrate the relationships between HIV-1 subtypes in Taiyuan and other regions due to the lack of corresponding reference sequences in the reference databases. The strains from Taiyuan were observed to be distributed across various clades in the neighbor-joining trees. Some strains formed a local cluster within the Taiyuan area, while immigrants and locals were not segregated into distinct clusters, but rather intertwined, indicating that there was a complex relationship of HIV-1 transmission between Taiyuan and other regions. The result highlighted that the principal driving force of the HIV/AIDS epidemic in Taiyuan was local infection. This result emphasizes the importance of localized prevention strategies to help refine or devise tailored interventions.

In this study, through a transmission network based on the links of different HIV-1 clusters among various risk groups (Figure 4 and Figure 5), the transmission linkages among various risk groups showed significant differences. Molecular network analysis showed more connections between the route of MSM and heterosexual transmission to be slightly higher than the route of MSM, indicating the association between the two routes of heterosexual transmission and MSM. Within the CRF07_BC_Cluster1 and CRF01_AE _Cluster5 network, the correlation among MSM was the highest, while in other clusters, heterosexual individuals showed stronger correlations, indicating diverse “sources” of the different HIV-1 clusters. Moreover, our molecular network analysis revealed that within some transmission clusters, diverse HIV transmission routes coexist within the same network. Interestingly, a “key person" emerged in various HIV transmission routes: individuals who have sex with men but identify as heterosexual. A possible explanation for this might be that some MSM with newly diagnosed HIV-1 infection may not disclose their true route of transmission during epidemiological surveys due to stigma and discrimination, as well as to the fact that they may also engage in sexual activities with women. The unreliability of self-reported data regarding acquired sexual behaviors in routine HIV/AIDS epidemiological surveys may therefore impact the formulation of effective control measures. To better understand HIV transmission routes, it is essential to employ advanced epidemiological methods, including the collection of detailed transmission information, including the occurrence of engaging in acts such as oral, vaginal, or anal sex. Collaborative efforts, combining molecular network analysis with enhanced field epidemiological surveys, can quantify local HIV transmission. Further research will enable us to identify and address discrepancies, resulting in a clearer understanding of the HIV/AIDS epidemic patterns in Taiyuan and providing valuable insights for prevention and treatment strategies.

Upon further analysis of the molecular network, we found that individuals in clusters were more likely to be household-registered outside of Taiyuan. Migration has been identified as a major factor driving the spread of the HIV epidemic across nations [50]. The uneven economic development in Taiyuan may prompt some of the population to seek better employment opportunities and living conditions in other urban centers. These individuals, flowing between their workplace and home, brining not only money, but also HIV-1, along with their higher levels of sexual risk, including their likelihood of participating in unprotected sex [51], could potentially act as a bridge, facilitating viral transmission from other provinces or cities to their home regions.

CRF79_0107 was identified among MSM in Jincheng and Datong cities, Shanxi Province, in 2015 [52]. Subsequently, this subtype was also detected in Taiyuan City in 2016, becoming the second largest cluster in the molecular transmission network, thereby playing a crucial role in the transmission process [42]. This subtype has also been detected in other regions of China besides Shanxi Province. For instance, it was recently identified in Hangzhou in 2019, where nine cases were found among 857 amplified samples, involving MSM individuals across different age groups [29]. Similarly, in April 2019, five cases were identified from 1297 amplified samples in Sichuan Province, all linked to MSM individuals [53]. Interestingly, this research reveals that CRF79_0107 can infect women, which is a significant finding. These findings suggest that the spread of CRF79_0107 has extended beyond the MSM individuals to other sexually active populations, indicating a clear provincial-level transmission pattern. The primary focus should be on the comprehensive understanding of the epidemiological characteristics of this subtype clustering, pinpointing key transmission groups, and implementing targeted prevention and control measures.

Notably, molecular network analysis revealed that Cluster7 comprises two CRF08_BC sequences and four URFs. Both neighbor-joining phylogenetic tree analysis and molecular network analysis indicate that these four URFs exhibit similarities with CRF08_BC. Based on the analysis, we tend to classify these four URFs as CRF08_BC. However, upon analysis of the average differences in nucleotides and amino acids, we conclude that URF exhibits a closer relationship with the CRF113_0107 subtype. Therefore, further gene sequencing, as well as functional studies, are needed to verify the association. Furthermore, a larger cluster was formed by CRF107_01B, initially detected in the MSM population in Heilongjiang [54]. We hypothesize that the spread of this subtype to Taiyuan was facilitated by population mobility.

Our study offered insights into the local transmission characteristics and epidemic patterns of HIV-1 subtypes within various high-risk behavior groups in Taiyuan. Nevertheless, it has several limitations. Firstly, the molecular network deduced from HIV-1 *pol* sequences represents only a fraction of the comprehensive local risk behavior network, excluding unsequenced and undiagnosed individuals. Secondly, epidemiological data, particularly in regards to sexual contact methods, rely solely on individual reports, potentially introducing information bias.

## 5. Conclusions

Our study is the first to apply a detailed molecular epidemiological approach to better explore the local transmission characteristics and epidemic pattern of HIV-1 subtypes among various sexual risk groups in Taiyuan City. Our findings emphasize that it is necessary to conduct in-depth research and precise intervention targeting key clusters/individuals, exploring new models based on the HIV-1 molecular transmission network, and implementing measures such as HIV/AIDS detection and exposure prevention to effectively block the ongoing transmission of HIV/AIDS and reduce the incidence of new infections. Understanding these epidemic dynamics in real time is of increasing importance for public health management in terms of guiding prevention efforts.

## Figures and Tables

**Figure 1 viruses-16-01174-f001:**
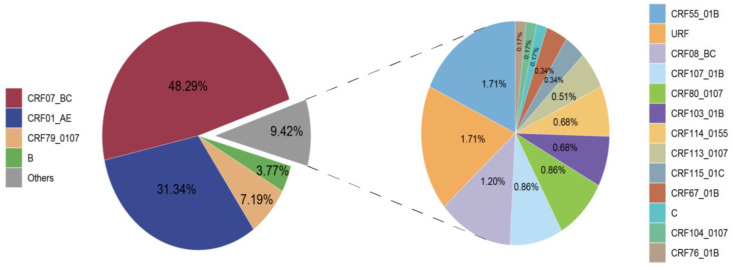
Distribution of HIV-1 subtypes in Taiyuan.

**Figure 2 viruses-16-01174-f002:**
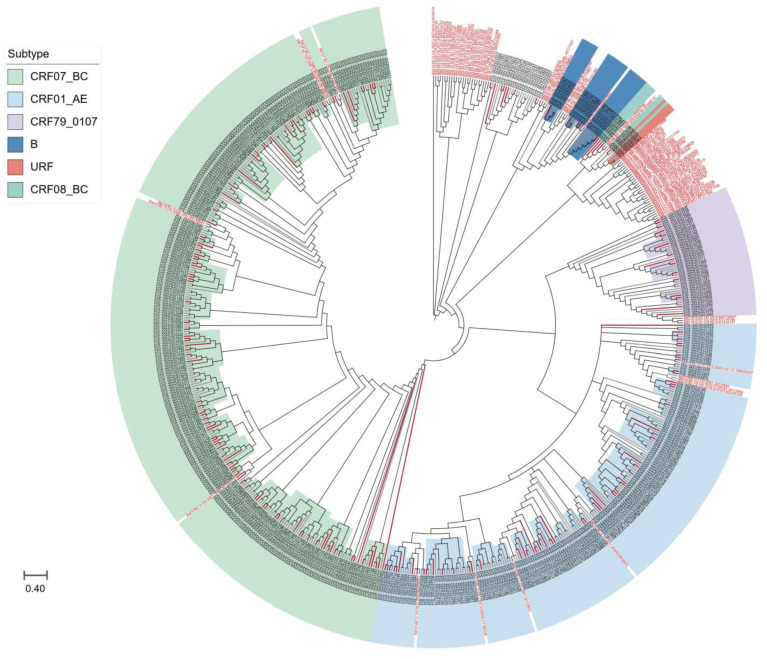
The neighbor-joining phylogenetic tree of HIV-1 *pol* sequences obtained from Taiyuan. Different colors represented different subtypes; red branches represent immigrant sequences, and black branches show local sequences; the reference sequences are indicated in red font.

**Figure 3 viruses-16-01174-f003:**
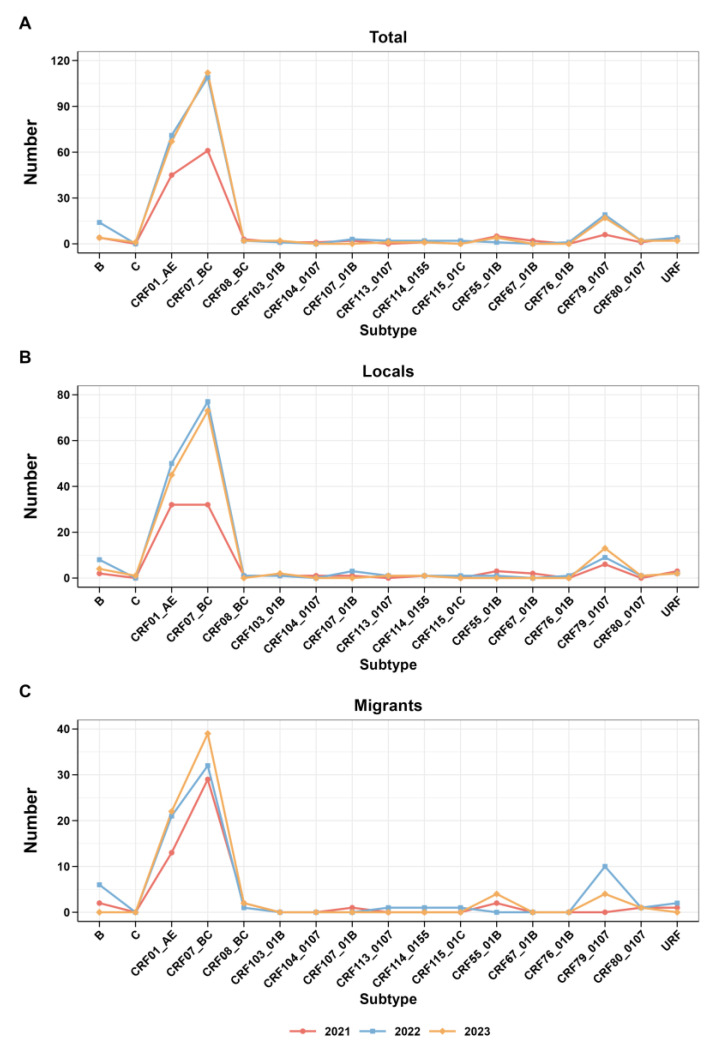
Trends in the subtypes of HIV-1 from 2021 to 2022. (**A**) Trends of total; (**B**) trends of locals; (**C**) trends of migrants.

**Figure 4 viruses-16-01174-f004:**
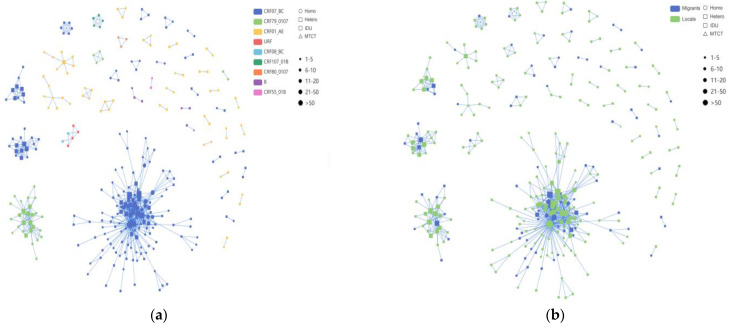
The molecular transmission; network diagram: (**a**) HIV-1 molecular clusters coded by subtypes; (**b**) HIV-1 molecular clusters coded by household registration level. Individuals infected with HIV-1 through homosexual contact are labeled with circles (○), individuals infected with HIV-1 through heterosexual contact are labeled with rectangles (□) and individuals infected with HIV-1 through injection drug use are labeled with rounded rectangles (
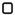
); this study comprises two cases, both belonging to Cluster1. Individuals infected with HIV-1 through mother-to-child transmission are labeled with upward-pointing triangles (△). Migrant subjects are shown in blue, and locals are indicated in green, respectively. Individuals infected with different HIV-1 subtypes are displayed in different colors.

**Figure 5 viruses-16-01174-f005:**
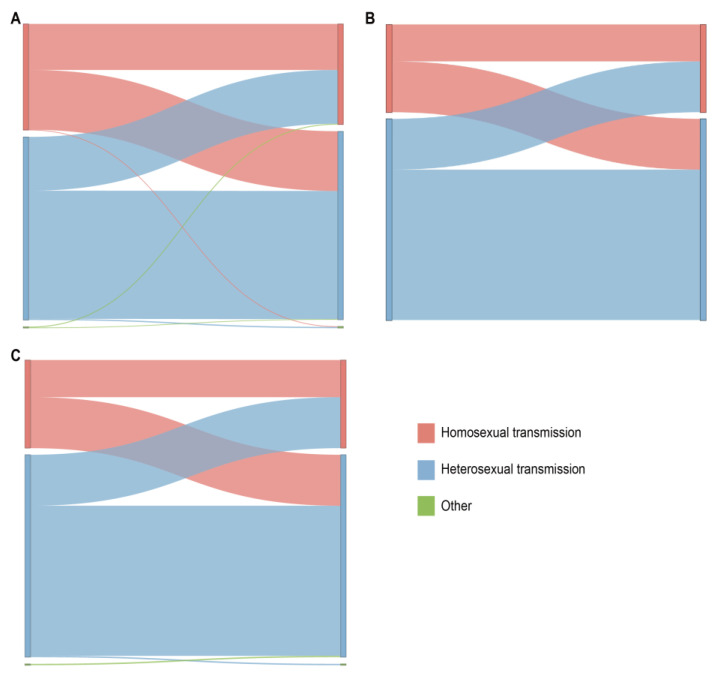
Linkage analysis of different risk behavior groups within the main HIV-1 subtypes in the network. The color indicates the different sexual contact risk groups. (**A**) Sankey diagram of CRF07_BC. (**B**) Sankey diagram of CRF01_AE. (**C**) Sankey diagram of other HIV-1 subtypes.

**Table 1 viruses-16-01174-t001:** Data of newly diagnosed HIV-infected individuals in Taiyuan.

Variables	Frequency (%)
**Total**	720 (100.0)
**Gender**	
Male	653 (90.7)
Female	67 (9.3)
**Ethnicity**	
Han	711 (98.8)
Others	9 (1.3)
**Age group (years)**	
<30	178 (24.7)
30–39	221 (30.7)
40–49	121 (16.8)
≥50	200 (27.8)
**Marital status**	
Single	354 (49.2)
Married	100 (13.9)
Divorced or widowed	266 (36.9)
**Education**	
Junior high school and below	261 (36.3)
High school or technical secondary school	160 (22.2)
College and above	299 (41.5)
**Household registration**	
Locals ^1^	459 (63.8)
Migrants ^2^	261 (36.2)
**Occupation**	
Workers	162 (22.5)
Farmers	112 (15.6)
Domestic workers and unemployed individuals	297 (41.3)
Commercial service workers	34 (4.7)
Students	27 (3.8)
Others	88 (12.2)
**Transmission route**	
Homo ^3^	261 (36.3)
Hetero ^4^	447 (62.1)
Others	12 (1.7)
**Sampling Region**	
Xiaodian District	195 (27.1)
Yingze District	93 (12.9)
Wanbailin District	162 (22.5)
Xinghualing District	109 (15.1)
Jiancaoping District	71 (9.9)
Jinyuan District	27 (3.8)
Gujiao City	18 (2.5)
Qingxu County	25 (3.5)
Loufan County	12 (1.7)
Yangqu County	8 (1.1)
**Sampling time**	
2021	168 (23.3)
2022	282 (39.1)
2021	270 (37.5)

^1^ Locals: individuals with household registered in Taiyuan; ^2^ migrants: individuals with household registered outside Taiyuan; ^3^ homo: homosexual transmission; ^4^ hetero: heterosexual transmission.

**Table 2 viruses-16-01174-t002:** General distribution of epidemic subtypes in Taiyuan.

Variables	Total [%] ^a^	The Frequency and Prevalence of Different HIV-1 Subtypes (%) ^b^						
	N = 584	CRF01_AE	CRF07_BC	CRF79_0107	B	Others *	χ^2^	*p* Value
**Gender**							11.119	0.025
Male	529 [90.6]	173(32.7)	246(46.5)	40(7.6)	18(3.4)	52(9.8)		
Female	55 [9.4]	10(18.2)	36(65.5)	2(3.6)	4(7.3)	3(5.5)		
**Ethnicity**							2.955	0.565
Han	576 [98.6]	181(31.4)	276(47.9)	42(7.3)	22(3.8)	55(9.5)		
Others	8 [1.4]	2(25.0)	6(75.0)	0(0.0)	0(0.0)	0(0.0)		
**Age group (years)**							18.277	0.108
<30	145 [24.8]	47(32.4)	64(44.1)	9(6.2)	5(3.4)	20(13.8)		
30–39	180 [30.8]	67(37.2)	76(42.2)	18(10.0)	7(3.9)	12(6.7)		
40–49	89 [15.2]	23(25.8)	45(50.6)	8(9.0)	4(4.5)	9(10.1)		
≥50	170 [29.1]	46(27.1)	97(57.1)	7(4.1)	6(3.5)	14(8.2)		
**Marital status**							9.915	0.271
Single	286 [49.0]	102(35.7)	124(43.4)	23(8.0)	10(3.5)	27(9.4)		
Married	75 [12.8]	26(34.7)	37(49.3)	3(4.0)	3(4.0)	6(8.0)		
Divorced or widowed	223 [38.2]	55(24.7)	121(54.3)	16(7.2)	9(4.0)	22(9.9)		
**Education**							18.033	0.021
Junior high school and below	201 [34.4]	52(25.9)	112(55.7)	12(6.0)	12(6.0)	13(6.5)		
High school or technical secondary school	138 [23.6]	47(34.1)	56(40.6)	10(7.2)	5(3.6)	20(14.5)		
College and above	245 [42.0]	84(34.3)	114(46.5)	20(8.2)	5(2.0)	22(9.0)		
**Household registration**							1.699	0.791
Locals	385 [65.9]	127(33.0)	182(47.3)	28(7.3)	14(3.6)	34(8.8)		
Migrants	199 [34.1]	56(28.1)	100(50.3)	14(7.0)	8(4.0)	21(10.6)		
**Occupation**							27.864	0.113
Workers	140 [24.0]	37(26.4)	68(48.6)	9(6.4)	9(6.4)	17(12.1)		
Farmers	91 [15.6]	25(27.5)	54(59.3)	6(6.6)	2(2.2)	4(4.4)		
Domestic workers and unemployed individuals	228 [39.0]	72(31.6)	108(47.4)	19(8.3)	9(3.9)	20(8.8)		
Commercial service workers	29 [5.0]	7(24.1)	13(44.8)	4(13.8)	0(0.0)	5(17.2)		
Students	23 [3.9]	9(39.1)	9(39.1)	0(0.0)	1(4.3)	4(17.4)		
Others	73 [12.5]	33(45.2)	30(41.1)	4(5.5)	1(1.4)	5(6.8)		
**Transmission route**							4.369	0.822
Homosexual behavior	208 [35.6]	68(32.7)	97(46.6)	17(8.2)	5(2.4)	21(10.1)		
Heterosexual behavior	365 [62.5]	112(30.7)	179(49.0)	24(6.6)	16(4.4)	34(9.3)		
Others	11 [1.9]	3(27.3)	6(54.5)	1(9.1)	1(9.1)	0(0.0)		
**Sampling time**							14.191	0.077
2021	136 [23.3]	45(33.1)	61(44.9)	6(4.4)	4(2.9)	20(14.7)		
2022	233 [39.9]	71(30.5)	109(46.8)	19(8.2)	14(6.0)	20(8.6)		
2023	215 [36.8]	67(31.2)	112(52.1)	17(7.9)	4(1.9)	15(7.0)		

^a^ Numbers in square brackets show the proportion of the cases as a percentage of the total 584 subjects. ^b^ Numbers in parentheses show the proportion of HIV-1 subtypes as a percentage of each variable. * Other subtypes including CRF55_01B (1.71%), URF (1.71%), CRF08_BC (1.20%), CRF107_01B (0.86%), CRF80_0107 (0.86%), CRF103_01B (0.68%), CRF114_0155 (0.68%), CRF113_0107 (0.51%), CRF115_01C (0.34%), CRF67_01B (0.34%), subtype C (0.17%), CRF104_0107 (0.17%), and CRF76_01B (0.17%).

**Table 3 viruses-16-01174-t003:** Univariate and multivariate analysis of clustered and non-clustered subjects in the molecular network.

Variables	Sequences					Univariate Analysis			Multivariate Analysis
	Total (N = 584)	Non-Clustered (N = 271)	Clustered (N = 313)	χ^2^	*p* Value	OR * (95% CI)	*p* Value	AOR * (95% CI)	*p* Value
**Gender**				0.513	0.474				
Male	529 [90.6]	248 (91.5)	281 (89.8)			1.000			
Female	55 [9.4]	23 (8.5)	32 (10.2)			0.814 (0.464–1.429)	0.474		
**Ethnicity**				2.667	0.102				
Han	576 [98.6]	265 (97.8)	311 (99.4)			1.000			
Others	8 [1.4]	6 (2.2)	2 (0.6)			3.521 (0.705–17.590)	0.125		
**Age group (years)**				3.654	0.301				
<30	145 [24.8]	74 (27.3)	71 (22.7)			1.000	0.302		
30–39	180 [30.8]	88 (32.5)	92 (29.4)			0.705 (0.451–1.101)	0.124		
40–49	89 [15.2]	37 (13.7)	52 (16.6)			0.768 (0.504–1.171)	0.220		
≥50	170 [29.1]	72 (26.6)	98 (31.3)			1.033 (0.614–1.737)	0.904		
**Marital status**				1.641	0.440				
Single	286 [49.0]	139 (51.3)	147 (47.0)			1.000	0.441		
Married	75 [12.8]	36 (13.3)	39 (12.5)			0.799 (0.562–1.137)	0.213		
Divorced or widowed	223 [38.2]	96 (35.4)	127 (40.6)			0.819 (0.484–1.384)	0.456		
**Education**				0.268	0.875				
Junior high school and below	201 [34.4]	96 (35.4)	105 (33.5)			1.000	0.875		
High school or technical secondary school	138 [23.6]	64 (23.6)	74 (23.6)			0.906 (0.623–1.317)	0.605		
College and above	245 [42.0]	111 (41.0)	134 (42.8)			0.927 (0.630–1.456)	0.840		
**Household registration**				11.841	0.001				
Locals	385 [65.9]	159 (58.7)	226 (72.2)			1.000			
Migrants	199 [34.1]	112 (41.3)	87 (27.8)			1.830 (1.295–2.586)	0.001	2.124 (1.457–3.095)	0.000
**Occupation**				2.687	0.748				
Workers	140 [24.0]	67 (24.7)	73 (23.3)			1.000	0.749		
Farmers	91 [15.6]	43 (15.9)	48 (15.3)			1.120 (0.636–1.972)	0.695		
Domestic workers and unemployed individuals	228 [39.0]	97 (35.8)	131 (41.9)			1.147 (0.619–2.125)	0.662		
Commercial service Workers	29 [5.0]	15 (5.5)	14 (4.5)			1.388 (0.818–2.355)	0.224		
Students	23 [3.9]	12 (4.4)	11 (3.5)			0.959 (0.406–2.269)	0.925		
Others	73 [12.5]	37 (13.7)	36 (11.5)			0.942 (0.369–2.407)	0.901		
**Transmission route**				3.489	0.175				
Homo	208 [35.6]	92 (33.9)	116 (37.1)			1.000	0.204		
Hetero	365 [62.5]	171 (63.2)	194 (62.0)			3.362 (0.867–13.033)	0.079		
Others	11 [1.9]	8 (3.0)	3 (1.0)			3.025 (0.790–11.586)	0.106		
**Sampling time**				4.891	0.087				
2021	136 [23.3]	69 (25.5)	67 (21.4)			1.000	0.087		
2022	233 [39.9]	115 (42.4)	118 (37.7)			0.660 (0.428–1.017)	0.060		
2023	215 [36.8]	87 (32.1)	128 (40.9)			0.697 (0.480–1.014)	0.059		
**Subtypes**				76.821	0.000				
CRF01_AE	183 [31.3]	123 (45.4)	60 (19.2)			1.000	0.000	1.000	0.000
CRF07_BC	282 [48.3]	87 (32.1)	195 (62.3)			0.903 (0.516–1.583)	0.722	0.856 (0.484–1.511)	0.591
CRF79_0107	42 [7.2]	11 (4.1)	31 (9.9)			4.151 (2.438–7.065)	0.000	4.270 (2.487–7.334)	0.000
Other subtypes	77 [13.2]	50 (18.5)	27 (8.6)			5.219 (2.271–11.993)	0.000	5.313 (2.284–12.358)	0.000

* AOR: adjusted odds ratio; OR: odds ratio.

**Table 4 viruses-16-01174-t004:** Characteristics of the large molecular transmission clusters.

Cluster No.	Subtypes	Nodes	Gender	Age	Transmission Route	Populations
Cluster1	CRF07_BC_N	133	M *–126, F *–7	G1 *–31, G2 *–41, G3 *–31, G4 *–30	Homo–58, Hetero–73, IDUs = 2	Locals–94, migrants–39
Cluster2	CRF79_0107	27	M–26, F–1	G1–4, G2–11, G3–7, G4–5	Homo–6, Hetero–21	Locals–19, migrants–8
Cluster3	CRF07_BC_O	16	M–13, F–3	G1–1, G3–1, G4–14	Homo–1, Hetero–15	Locals–11, migrants–5
Cluster4	CRF07_BC_O	10	M–3, F–7	G2–4, G3–1, G4–5	Hetero–10	Locals–7, migrants–3
Cluster5	CRF01_AE_C4	8	M–8	G1–1, G2–5, G3–2	Homo–5, Hetero–3	Locals–7, migrants–1
Cluster6	CRF01_AE_C4	8	M–8	G2–4, G4–4	Homo–3, Hetero–5	Locals–7, migrants–1
Cluster7	URF	6	M–6	G2–2, G4–4	Hetero–6	Locals–5, migrants–1
Cluster8	CRF07_BC_O	6	M–2, F–4	G4–6	Hetero–6	Locals–5, migrants–1
Cluster9	CRF107_01B	5	M–5	G1–2, G2–2, G3–1	Homo–3, Hetero–2	Locals–4, migrants–1

* M: male; F: female; G1: age group 1 (<30 years); G2: age group 2 (30–39 years); G3: age group 3 (40–49 years); G4: age group 4 (≥50 years). The number after the dash denotes the number of subjects (omitted if all in this category).

## Data Availability

The data are deposited in the National Microbiology Data Center (NMDC), with accession numbers NMDCN0003KKF-NMDCN0003L6E, NMDCN0003L8S-NMDCN0003L93 (https://nmdc.cn/resource/genomics/sequence). The datasets used and/or analyzed in this study are available from the corresponding author upon reasonable request.

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
