# Peer review of "Characteristics of Subtype and Molecular Transmission Networks among Newly Diagnosed HIV-1 Infections in Patients Residing in Taiyuan City, Shanxi Province, China, from 2021 to 2023"

_viruses, 2024, doi:10.3390/v16071174_

Round 1

Reviewer 1 Report

Comments and Suggestions for Authors

HIV-1 remains a health challenge worldwide in the 95-95-95 era. This is an interesting  paper describing a regional epidemic in China. In 2023, there were 1,289,700 people living with HIV in China and 110,491 new infections. The viral epidemic has diversified to include new viral recombinant forms spreading among drug user and MSM populations to heterosexual groups.

This study was designed to apply molecular network analyses to better characterize the introduction and spread of HIV-1 subtypes in Taiyuan, Shanaxi province over the 2021 to 2023 period. The paper is  well-written and the phylogenetic and Microbe Trace molecular network analyses are appropriate to deduce the clustering of viral sequences at a 1.5% genetic threshold.

The predominant epidemic in Taiyuan was among men (90%) with a significant migrant population (36%) and reported MSM (36%) and HET (62%) route of transmission.  Among the 20 HIV strains and circulating recombinant forms, there were 4 major subtypes

·         CRF07_BC: 48%    (50 yrs+ 47% college 49% HET)

·         CRF01_AE: 31%   (30% HET)

·         CRF079_0107: 7%

·         B: 4%

Multivariate analyses compared features of these four strains in 2023 (eg. Table 2)

Molecular network analyses may address the inherent weaknesses in classical epidemiology in describing the spread of different HIV subtype outbreaks since HIV-1 is stigmatized spreading in highly vulnerable and often marginalized stigmatized populations.  

My major suggestion would be to provide a summary Table or figure illustrating or quantifying the distribution of cluster sizes in each of these 4 predominant epidemics, the median and range.  Although data is presented for 2021-2023, is there any differences in recent temporal trends and migrant ratios that may impact clustering.  How significant is crossover among risk groups.  

Clustering designation (Yes vs. No) may not be the ideal strategy to resolve transmission dynamics based on depth of sampling and given the absence of large molecular clusters within any given subtype group.

As an example, the CRF 07 large cluster 1 of 133 is significantly associated with MSM. A well written paragraph (lines 287-301) provides a discussion on route of transmission.  Do the authors think that the preponderance of small clusters related to HET infections be a challenge in epidemic control? Given stigma, how confident are the authors to categorize MSM vs. HET and anal vs. HET route of transmission?  

Author Response

Comments 1: My major suggestion would be to provide a summary Table or figure illustrating or quantifying the distribution of cluster sizes in each of these 4 predominant epidemics, the median and range.

Response 1: Thanks for your comments. Our data are more appropriately expressed as rates. To our knowledge, frequency distribution and percentage were used to summarize categorical variables, and median with interquartile range (IQR) was used to describe continuous variables. Therefore, the categorical variables in our Table 2 should be expressed in terms of rates, and not medians and range.

Comments 2: Although data is presented for 2021-2023, is there any differences in recent temporal trends and migrant ratios that may impact clustering.

Response 2: Thanks for your comments. Revised as suggested [Page 9 Lines 231-237 and Figure 3], data from various years were incorporated into Table 1, Table 2, and Table 3, followed by subsequent data analysis.

Comments 3: How significant is crossover among risk groups.

Response 3: Thank for your comments. Revised as suggested. We have drawn Sankey diagrams based on the links of different HIV-1 subtypes among the various risk groups in the network, analysis of different risk behaviour groups with the main HIV-1 subtypes in the network [Page 13 Lines 290-296, Page15 Lines 361-364 and Figure 3].

Comments 4:  Do the authors think that the preponderance of small clusters related to HET infections be a challenge in epidemic control? Given stigma, how confident are the authors to categorize MSM vs. HET and anal vs. HET route of transmission?  

Response 4: Thank for your comment. We have re-written this part according to the Reviewers suggestion [Page 15 Line 375, 376, Page16 Lines 377-382].

We appreciate for your warm work earnestly, and hope that the correction will meet with approval. Once again, thank you very much for your comments and suggestions.

Reviewer 2 Report

Comments and Suggestions for Authors

The article "Characteristics of subtype and molecular transmission network among newly diagnosed HIV‐1 infections residing in Taiyuan City, Shanxi Province, China from 2021 to 2023" by Gao and colleagues summarizes and describes 584 HIV-1 pol sequences obtained in Taiyuan from an epidemiolgoical and molecular point of view. The article is clear and well written, and constitutes a small but important piece in the study of current HIV-1 evolution. I have however some concerns on the methodology, way of presenting results, and more in general on the depth of significance of their findings.

- Paragraph 3.2 I must say that the description of HIV-1 subtypes is lacking from a molecular standpoint. The authors do not describe the actual genomic sequence differences between strains, nor how these reflect into protein changes or even structural and functional changes. The authors should show how the subtypes of pol may reflect different functionality of the protein, in several ways: 1) a table listing all the aminoacid differences between strains 2) highlighting the position of mutation on the protein structure of pol (available in many entries of the Protein Data Bank) and 3) linking them to other HIV-1 sequencing articles obtained from outside Taiyuan.

- Figure 2. The phylogenetic trees are too large to show in a single figure. Even at high resolution, it is impossible to discerne the individual sequences or even to grasp the writing in the legend, associating colors to cluster. I suggest the authors dedicate a full page to their main phylogenetic diagram, and move the other two as new figures, or even as supplementary figures.

- Discussion. While rich of anecdotes about the patient group, the discussion fails to tie the findings of the authors with existing HIV-1 epidemiological and genomic studies outside of Taiyuan. Some questions remain unanswered: are the observed subtypes unique to the Taiyuan city? Is there some sort of evolutionary pressure determining the observed quasispeciation of HIV-1 in the study group? Can the observed new HIV-1 pol sequences generate proteins with different polymerase efficiency or proofreading capability? In other words, the authors should put a higher effort in transforming an observational study into a more insightful study.

Author Response

Comments 1: Paragraph 3.2 I must say that the description of HIV-1 subtypes is lacking from a molecular standpoint.

Response 1: Thanks for your comment. Revised as suggested [Page 1, Lines 40-44, Page 2, Lines 45-47], we described HIV-1 subtypes in more detail.

Comments 2: The authors do not describe the actual genomic sequence differences between strains, nor how these reflect into protein changes or even structural and functional changes, and a table listing all the amino acid differences between strains

Response 2: Thanks for your comment. Revised as suggested. We calculated the average differences in amino acids and nucleotides among various subtypes, as well as within them, the entire table can be found in the Supplementary Material [Supplementary Table S1].

Comments 3: Highlighting the position of mutation on the protein structure of pol (available in many entries of the Protein Data Bank). Is there some sort of evolutionary pressure determining the observed quasispeciation of HIV-1 in the study group? Can the observed new HIV-1 pol sequences generate proteins with different polymerase efficiency or proofreading capability?

Response 3: Thanks for your comment. Our article primarily delves into HIV-1 subtypes and gene characteristics, additionally, our subsequent investigation and exploration of protein-related issues will be elaborated upon in the subsequent article focused on drug resistance. Furthermore, to the best of our knowledge, we have yet to uncover any pertinent literature. Consequently, we humbly request your assistance, if possible, could you kindly recommend any relevant articles, which would be appreciated.

Comments 4: Linking them to other HIV-1 sequencing articles obtained from outside Taiyuan.

Response 4: Thanks for your comment. In the phylogenetic tree, reference strains from various regions and years were chosen for comparison with the sequences obtained in Taiyuan City from 2021 to 2023. The reference strains are highlighted in red font. Revised as suggested [Page5, Lines 180-182, Page6, Lines 183-185 and Figure 2].

Comments 5: Figure 2. The phylogenetic trees are too large to show in a single figure. Even at high resolution, it is impossible to discerne the individual sequences or even to grasp the writing in the legend, associating colors to cluster. I suggest the authors dedicate a full page to their main phylogenetic diagram, and move the other two as new figures, or even as supplementary figures.

Response 5: Thanks for your comment. We think this is an excellent suggestion. We have modified the figure using the suggestions and also introduced the entire phylogenetic tree in detail, and move the other two as supplementary figures. In addition, we also provide PDF format for clearer observation.

Comments 6: Are the observed subtypes unique to the Taiyuan city?

Response 6: Thanks for your comment. Revised as suggested. It has been added in the discussion part [Page 15, Lines 349-358].

Round 2

Reviewer 2 Report

Comments and Suggestions for Authors

The reviewers put real effort in improving their manuscript according to the suggestions, and even beyond that. The introduction (concernging the HIV-1 parts) is now more complete. The discussion (connecting the Taiyuan cases to the broad worldwide HIV-1 evolutionary scenario) is now more informative. I particularly appreciate the updated Figure 2 (the phylogentic tree) which is now not only readabale and informative of the HIV-1 strains detected, but also more graphically appealing.